# An Organizational-Level Workplace Intervention to Improve Medical Doctors’ Sustainable Employability: Study Protocol for a Participatory Action Research Study

**DOI:** 10.3390/ijerph21121561

**Published:** 2024-11-26

**Authors:** Anna van Duijnhoven, Juriena D. de Vries, Hanneke E. Hulst, Margot P. van der Doef

**Affiliations:** 1Health, Medical and Neuropsychology Unit, Leiden University, 2333 AK Leiden, The Netherlands; j.d.de.vries@fsw.leidenuniv.nl (J.D.d.V.); h.e.hulst@fsw.leidenuniv.nl (H.E.H.); doef@fsw.leidenuniv.nl (M.P.v.d.D.); 2Leiden Institute for Brain and Cognition, Leiden University, 2333 AK Leiden, The Netherlands

**Keywords:** sustainable employability, psychosocial safety climate, job characteristics, medical doctors, organizational-level workplace intervention, participatory action research, effect evaluation, process evaluation

## Abstract

Compromised Sustainable Employability (SE) of medical doctors is a concern for the viability of healthcare and, thus, for society as a whole. This study (preregistration: ISRCTN15232070) will assess the effect of a two-year organizational-level workplace intervention using a Participatory Action Research (PAR) approach on the primary outcome SE (i.e., burnout complaints, work engagement, and job satisfaction) and secondary outcomes (i.e., turnover intention, occupational self-efficacy, and perceived impact on health/well-being) in medical doctors. It will also examine whether changes in Psychosocial Safety Climate (PSC), job characteristics (i.e., job demands and resources), and perceived impact on the work situation mediate these effects, and which process factors (i.e., degree of actual implementation of changes, information provision, management support, medical doctors’ involvement, and mental models) are important to the intervention’s success. A pre-post design will be used, including 24 groups of medical doctors (approximately *N* = 650). Data will be collected at four measurement points (a pre-test, two intermediate evaluations, and a post-test) and analyzed using linear mixed-effect models. The results will provide insights into the effectiveness of the intervention in promoting SE and will inform future organizational-level workplace interventions about the mediators and factors in the implementation process that contribute to its effects.

## 1. Introduction

Sustainable employability (SE) is perceived as the ability of employees to function at work over time [1]. SE is a rising concern for society, the healthcare system, and especially among medical doctors. The healthcare system puts enormous pressure on medical staff to meet expectations and obligations (e.g., high targets and administrative requirements) [2]. This is especially tangible in current times of global healthcare staff shortages, estimated at around 15 million in 2020 and 10 million in 2030 [3]. Consequently, the employment and retention of medical doctors throughout their careers and the viability of healthcare are under threat [4]. Hence, interventions to preserve and improve SE within the medical healthcare sector are urgent.

This protocol paper outlines the design of a study evaluating an intervention aimed at preserving and improving three key SE indicators for medical doctors: burnout complaints, work engagement, and job satisfaction (based on [1], and referred studies). The first indicator, burnout complaints, is described as a state of emotional exhaustion, a sense of cynicism and detachment toward work, and reduced feelings of personal accomplishment and efficacy [5]. Burnout prevalence rates among medical doctors have reached high levels globally [4,6,7]. Approximately one in two U.S. medical doctors were found to fall into the risk category of at least one burnout complaint [6]. Furthermore, the U.S. costs of medical doctors’ burnout-related turnover, reduced clinical hours, and/or productivity were estimated at around USD 5000 to 10,000 per medical doctor per year [7] and USD 4.6 billion each year in total [8]. The second indicator, work engagement, is seen as a well-being outcome, characterized by vigor, dedication, and absorption related to one’s job [9,10]. Work engagement is important as it contributes to the job performance and well-being of employees [9,11]. The third and last indicator, job satisfaction, is a broad term for how satisfied employees are with their jobs and is seen as an indicator of occupational well-being [12,13,14]. Ensuring medical doctors’ SE is crucial, as compromised SE—manifested through high burn-out complaints, low work engagement, and job dissatisfaction—can have far-reaching effects. At the individual level, compromised SE impacts mental, physical, and psychological health, and quality of life [7,15,16,17,18,19]. At the organizational level, it negatively influences organizational commitment, job and career satisfaction, productivity, and performance [15,18,19,20,21,22,23,24,25]. Moreover, it directly impacts the quality of care provided [7,20,22,26,27]. Given the serious implications of medical doctors’ compromised SE, it is crucial to develop and evaluate interventions aimed at enhancing SE.

### 1.1. Theoretical Background

Research has shown that interventions that are based on the job demands–resources model (JD-R model) [9,28,29,30,31] are promising to enhance the SE of medical doctors [32,33]. The JD-R model is one of the currently most researched occupational stress models. Within the JD-R model, job characteristics are considered essential risk or protective factors of SE [9,34]. Job characteristics entail job demands that require physical and/or psychological effort (health impairment process), and job resources that stimulate the achievement of goals, and personal growth, and potentially offset the negative effect of high job demands (motivational process). The combination of excessive job demands and low job resources, increases the risk of burnout, whereas (not too) high job demands and favorable job resources stimulate work engagement and job satisfaction [34,35]. More recently, the JD-R model has been extended with Psychosocial Safety Climate (PSC) [28,29,30]. PSC refers to organizational values and actions aimed at protecting employees’ psychological health, well-being, and safety [28]. PSC is seen as “a cause of causes”, influencing job characteristics, and through that pathway decreases the risk of burnout complaints and enhances work engagement [28,29]. Hence, to develop effective interventions to enhance SE, the intervention must aim to improve both PSC and job characteristics [28,29,30,31,36,37,38,39].

Certain job characteristics are strongly associated with compromised SE among medical doctors. In the current healthcare sector, medical doctors frequently face high job demands, such as emotional and psychological stressors, time pressure, long and excessive working hours, high workload, and lack of time for administrative tasks [2]. These demands may compromise SE [2,7,40]. Job resources, for example, autonomy at work, constructive feedback, rewards, development opportunities, and social support from colleagues and management, are important to allow medical doctors to cope with these high job demands in support of SE. However, research indicates that the availability of those resources is also compromised in the current healthcare sector [2,7,16,17,22,41,42,43]. To date, the relationships between job characteristics and medical doctors’ burnout have been examined more extensively than those with job satisfaction and work engagement. Furthermore, many studies have employed a cross-sectional design (e.g., [7]). This highlights the need for stronger research designs, including a more comprehensive measure of SE (i.e., burnout complaints, work engagement, and job satisfaction) in this sample.

### 1.2. Organizational-Level Workplace Interventions

Organizational-level workplace interventions that address organizational factors and directly target PSC and job characteristics are considered the optimal approach for preserving and maintaining SE [44,45,46,47,48]. These interventions focus on structural improvement by changing the organization and management of work. In contrast, individual-directed interventions aim to enhance employees’ skills, resilience, and capacities to cope with problems (e.g., emotion regulation, stress management, and communication skills) [4]. As organizational-level workplace interventions address the main and structural causes of compromised SE, they have a higher potential to enhance and maintain SE than individualized solutions [4,32,33,45,49,50,51]. Indeed, organizational-level workplace interventions among medical doctors are effective in reducing the risk of burnout complaints and/or improving job satisfaction [32,33,52]. More specifically, a meta-analysis [33] demonstrated a medium decrease in medical doctors’ burnout complaints based on 7 incorporated studies. Similarly, a systematic review [32] demonstrated that 70 percent of the 50 incorporated studies presented organizational-level workplace interventions that proved to be effective in reducing medical doctors’ burnout complaints or stress, and/or increasing job satisfaction. The included interventions focused on improving teamwork, time (e.g., schedule adjustments or lowering excessive working hours), transitions (e.g., workflow changes), and/or technology. Consistent with these findings, another systematic review [52] also demonstrated the effectiveness of organizational-level workplace interventions on medical doctors’ burnout complaints and/or job satisfaction, specifically aimed at optimizing the digital environment for medical doctors. Despite the demonstrated success of organizational-level workplace interventions, previous studies were mainly focused on burnout, a few on job satisfaction, and rarely examined work engagement. This is unfortunate, as high levels of work engagement have sustainable effects on the health and well-being of employees [9,11]. Furthermore, previous studies rarely examined the mechanisms by which the intervention led to enhanced SE. That is, it remains unclear whether organizational-level workplace interventions enhanced SE through more favorable PSC and/or job characteristics. Insight into mechanisms of interventions is needed to refine theories and maximize the effectiveness of future interventions [46]. Thus, organizational-level workplace interventions for medical doctors aimed at enhancing all three SE indicators and their potential working mechanisms are required.

Whereas organizational-level workplace interventions are considered the best option to enhance SE, it is important that their recipients actively take part in such interventions to guarantee their success [46,53,54,55]. Participatory Action Research (PAR) favorably engages employees in various intervention phases (i.e., planning, implementing, and evaluating) [54]. PAR is a way of increasing the intervention appropriateness to a specific organizational context using employees’ and organizations’ resources and expertise. In this way, PAR tailors the intervention to employees’ and organizational needs and thereby optimizes its effectiveness [46,53,54]. Furthermore, PAR enhances employees’ autonomy, empowerment, and learning ability, by involving them as active agents in the intervention development and implementation, engaging them in discussing and determining the addressed problems, designing action plans, and implementing strategies to tackle these problems [46,53,54,55]. This joint ownership is an essential element of PAR, as it enhances shared responsibility and equips employees to address SE beyond the intervention period. In turn, joint ownership prevents the delegation of responsibility to external intervention providers (coaches) and avoids stagnation of the process once the intervention has finished.

Process evaluations are needed to understand how organizational-level workplace interventions are implemented in real-world settings. Common factors that influence the implementation process are the extent to which planned changes are implemented, (unexpected) organizational circumstances that take place during the intervention, employees’ involvement and participation, employees’ communication and information provision about the progress of the intervention, employees’ mental models (e.g., expectations), and management support [45,46,47,48,53,56]. A relevant example is a study [57] that implemented an organizational health intervention using a PAR approach in nursing ward teams. The study demonstrated that direct participants who were actively involved in various intervention phases (e.g., setting goals and action planning) experienced more positive intervention outcomes than indirect participants, who were represented by direct participants. Furthermore, the findings highlighted the importance of a successful implementation process, as indirect participants showed positive intervention outcomes in teams with a successful intervention implementation. Thus, a process evaluation of factors that could influence the intervention’s effectiveness, i.e., examining what works for whom, how (mechanisms), and under which circumstances (context), is needed to provide a complete picture of the effectiveness of organizational-level workplace interventions with a PAR approach [58].

To date, effect evaluations of organizational-level workplace interventions using a PAR approach among medical doctors are scarce. Two studies [59,60] did demonstrate the success of such interventions in improving aspects of care coordination among medical doctors, such as improved collaboration and communication between medical doctors, and management coordination. Furthermore, a process evaluation of such interventions is seldom conducted. One study [61] identified key implementation success factors (i.e., institutional support and medical doctors’ and managers’ willingness to participate). The current study is novel, as there are no existing studies that assess the effectiveness and process of an organizational-level workplace intervention using a PAR approach, to improve PSC and job characteristics, and through that pathway SE indicators.

Therefore, to fill this research gap, this protocol paper presents the design of a study aimed at testing a two-year-long organizational-level workplace intervention using a PAR approach to enhance medical doctors’ SE (see Figure 1). The groups are guided by process facilitators (coaches), but the medical doctors are the key agents of change in the intervention. We will evaluate the overall effect of the intervention on PSC, job characteristics (i.e., job demands and job resources), and the primary outcomes SE indicators (i.e., burnout complaints, work engagement, and job satisfaction). Additionally, we will examine the potential mediating role of job characteristics and PSC, and the facilitating and inhibiting factors during the implementation process of the intervention. In addition to the primary outcomes, we will examine the effect of the intervention on secondary outcomes highly relevant for medical doctors: turnover intention, occupational self-efficacy (OSE), and perceived impact on health/well-being. To specify, compromised SE can lead to turnover, which has a large negative effect on the healthcare system [7,16,19,20]. Moreover, OSE can be seen as a related SE indicator that captures medical doctors’ competence beliefs (based on [1]), that is, employees’ beliefs about mastering skills to succeed in tasks or challenges that are required by their job [62,63,64]. Further, the intervention has the potential to improve various health and well-being indicators, of which SE, turnover intention, and OSE are currently studied. To gauge an overall measure of potential improvements in health/well-being, medical doctors’ perceived impact on health/well-being is also being measured. Similarly, in addition to PSC and job characteristics as proposed mechanisms of the intervention, we will examine medical doctors’ perceived impact on the work situation. The reason for this is that the work situation is complex and comprehensive, encompassing a diversity of job characteristics. Although this study measures job characteristics extensively, not all potential changes in the work situation are captured by the assessed job characteristics.

### 1.3. Aim and Objectives

The overall aim of the study, as presented in this protocol paper, is to conduct an effect and process evaluation of an organizational-level workplace intervention using a PAR approach among medical doctors. The first objective is to evaluate the effectiveness of an organizational-level workplace intervention on three relevant SE indicators (burn-out complaints, work engagement, and job satisfaction; primary outcomes), turnover intention, OSE, and perceived impact on health/well-being (secondary outcomes). Previous research demonstrated that organizational-level workplace interventions are effective in improving medical doctors’ SE [32,33,52]. Based on previous research we expect:

**Hypothesis 1a.** 
*The organizational-level workplace intervention will improve the primary outcomes of burnout complaints, work engagement, and job satisfaction.*


**Hypothesis 1b.** 
*The organizational-level workplace intervention will improve the secondary outcomes of turnover intention, OSE, and perceived impact on health/well-being.*


The second objective is to assess whether changes in PSC, job characteristics (job demands and job resources), and perceived impact on the work situation, mediate the effect of the intervention on the primary and secondary outcomes. The JD-R model posits that the work environment predicts SE; an idea that has received extensive empirical support (e.g., [9]). Further, prior research indicates that organizational-level workplace interventions can directly improve the work environment (e.g., [46]). Based on these findings, it can be inferred that our organizational-level intervention enhances SE indicators through an improved work environment:

**Hypothesis 2a.** 
*The effect of the organizational-level workplace intervention on the primary outcomes of burnout complaints, work engagement, and job satisfaction is mediated by enhanced PSC, decreased job demands, enhanced job resources, and positive perceived impact on the work situation.*


**Hypothesis 2b.** 
*The effect of the organizational-level workplace intervention on the secondary outcomes of turnover intention, OSE, and perceived impact on health/well-being is mediated by enhanced PSC, decreased job demands, enhanced job resources, and positive perceived impact on the work situation.*


The last objective is to evaluate which process factors play an important role in the effectiveness of this organizational-level workplace intervention in improving the mediators, and the primary and secondary outcomes. Previous research highlights the importance of studying process factors that influence the success of organizational-level workplace interventions (e.g., [56]). We will examine various process factors known to impact the effectiveness of these interventions (e.g., [48]). We expect:

**Hypothesis 3a.** 
*The effect of the intervention on the mediators PSC, job characteristics, and perceived impact on the work situation will be stronger in groups of medical doctors with a more favorable intervention process: a higher degree of actual implementation of changes, better information provision, higher medical doctors’ involvement, more management support, and more favorable medical doctors’ mental models (appraisals of the focus and approach of the intervention, and positive expectations).*


**Hypothesis 3b.** 
*The effect of the intervention on the primary outcomes of burnout complaints, work engagement, and job satisfaction will be stronger in groups of medical doctors with a more favorable intervention process.*


**Hypothesis 3c.** 
*The effect of the intervention on the secondary outcomes turnover intention, OSE, and perceived impact on health/well-being will be stronger in groups of medical doctors with a more favorable intervention process.*


## 2. Materials and Methods

### 2.1. Study Design

This study is a pre-post intervention group design involving groups of medical doctors. No control group is included, given the difficulties of conducting a randomized experimental design in organizational-level workplace interventions [53]. Such interventions interact with contextual factors, consequently, it is difficult to maintain the context constant. Furthermore, a wait-list design is less feasible, because then the medical doctors have to be assessed for two years without receiving intervention.

### 2.2. Ethical Consideration and Trial Registration

This study has been approved by the Psychology Research Ethics Committee of Leiden University (registration numbers: 2020-09-29, V2-2611; 2023-04-04, V3-4509). Furthermore, this study is registered in the ISRCTN registry (registration number: ISRCTN15232070).

### 2.3. Study Setting, Sample and Recruitment

This study involves an organizational-level workplace intervention implemented by the Dutch Association of Salaried Doctors (LAD), among groups of medical doctors in diverse Dutch healthcare settings (e.g., hospitals and municipal health services). The inclusion criteria are that the group consists of medical doctors employed in a Dutch healthcare setting and that all group members support participating in the intervention. No specific exclusion criteria are applied. The aim is to recruit 24 groups that will participate until the end of the intervention and final evaluation. Groups subscribed themselves for the intervention, so there is no random selection of groups. The enrolment date varies per group, resulting in different start dates for the intervention across groups. Groups started between October 2020 and January 2023, and the last group is expected to finish in December 2024. It is important to note that the COVID-19 pandemic played a significant role during the intervention period, particularly between 2020 and mid-2022 in the Netherlands. COVID-19 had a significant impact on healthcare, the work environment of medical doctors, and potentially their SE [65].

### 2.4. The Organizational-Level Workplace Intervention

The intervention is a two-year-long organizational-level workplace intervention with the PAR approach, implemented by the LAD. The main goal is to enhance SE via improvements in PSC and job characteristics. Process facilitators (coaches) from the LAD guide and support the medical doctor groups throughout the intervention. The level of guidance offered by the process facilitators gradually diminishes as the intervention progresses. This ensures that the groups can independently sustain the change processes related to PSC and job characteristics once the intervention is finished. The intervention is divided into three distinctive phases, each defined by the level of guidance (see Figure 2): phase one with intensive guidance (8 months), phase two with guidance and support according to needs (6 months), and phase three with guidance as needed and necessary (10 months).

The study measurements serve as the starting point of the intervention and provide intermediate input for the process, in line with the PAR approach (see Figure 2). As such, the intervention starts with the pre-test, and the results are presented to the medical doctor group during a kick-off meeting by the researchers. The results identify the main problems regarding PSC, job characteristics, SE, turnover intention, and OSE. The group, as the active agent in the intervention, decides on the themes they wish to address, forming workgroups for each chosen theme. Each workgroup defines goals, develops action plans and implementation strategies, and initiates changes, under the guidance of the process facilitator. The process facilitator’s role is to guide and facilitate group discussions, ensuring that all medical doctors are heard and actively involved, providing relevant information, and helping to develop the skills and knowledge needed for a successful intervention process. Particularly, the process facilitator encourages the group to recognize their role as key agents in the intervention and supports them in taking ownership of this role. The process facilitator assists the group in goal-setting, identifying obstacles, finding solutions, implementing changes, evaluating progress, adjusting strategies, and ensuring the sustainability of the intervention. Plenary meetings are held with the entire medical doctor group, to stimulate collaboration, communication, and a coordinated and streamlined process among the various workgroups.

The meetings are intensively guided during intervention phase one. Intervention phase one ends with the first intermediate evaluation meeting, where researchers present intermediate feedback to the group about the course of the intervention process. The results are based on the first intermediate evaluation and additional online semi-structured interviews with a few medical doctors in the group and the process facilitator. The results highlight which identified problems have been solved, which require further improvement, and the facilitators (what is proceeding well) and barriers (how to improve the process) in the implementation process. The group uses this feedback to continue with intervention phase two. The level of guidance decreases in this phase (i.e., guidance and support according to needs). Phase two also ends with an intermediate evaluation meeting, during which the results of the second intermediate evaluation and online semi-structured interviews are presented by researchers to the group. Based on this feedback, the group continues with the last phase of the intervention: (further) development and implementation of changes. The level of guidance decreases again (i.e., guidance as needed and necessary), to equip the group to continue with the change process to maintain or improve SE beyond the intervention period. The intervention ends with the post-test and online semi-structured interviews of which the results are presented during a final meeting. This provides the group with insights into their achievements during the intervention and highlights themes that still require (further) attention in the future.

### 2.5. Procedure

This study includes four measurement points (T1, T2, T3, and T4) corresponding to the three distinctive phases of the intervention (see Figure 2). The first measurement point (T1) is the pre-test at the start of intervention phase one, assessing the primary and secondary outcomes, and mediators, through the online questionnaire. The pre-test (T1) is followed by three evaluation measurement points (T2, T3, and T4). The second measurement point (T2) is the first intermediate evaluation at the start of phase two (8 months), the third measurement point (T3) is the second intermediate evaluation at the start of phase three (14 months), and the fourth measurement point (T4) is the post-test at the end of phase three (23 months). At T2, T3, and T4, questionnaires assess the primary and secondary outcomes, mediators, and process factors (facilitators and barriers). In addition to the questionnaires, online semi-structured interviews are conducted with a few medical doctors in a group and the process facilitator. These interviews provide additional insights into barriers and facilitators regarding the intervention process.

### 2.6. Measures

#### 2.6.1. Online Questionnaires

The data are collected through online questionnaires, using the program Qualtrics. At the pre-test, the primary outcomes (burnout complaints, work engagement, and job satisfaction), secondary outcomes (turnover intention and OSE), and mediators (PSC and job characteristics) are measured. The intermediate evaluations and post-test include pre-test items, the mediator perceived impact on the work situation, the secondary outcome perceived impact on health/well-being, and process factors. See Table 1 for an overview of the measurement points of the primary and secondary outcomes, mediators, and process factors. It takes approximately 30 min to complete each questionnaire. To enhance the response rate, each questionnaire is available online for approximately three to four weeks, and multiple reminders are sent out.

#### 2.6.2. Primary Outcome Measures

The primary outcome is *SE*, which is assessed with three indicators: burnout complaints, work engagement, and job satisfaction.

##### Burnout Complaints

First, burnout complaints are measured by the work-related version of the Burnout Assessment Tool (BAT) [66]. BAT includes 23 items measuring four core burnout complaints, answered on a 5-point Likert scale (1 = never, 5 = always): emotional exhaustion (e.g., “At work, I feel mentally exhausted”), mental distance (e.g., “I feel a strong aversion towards my job”), cognitive impairment (e.g., “At work I struggle to think clearly”), and emotional impairment (e.g., “At work, I feel unable to control my emotions”). An average overall burnout score is calculated based on the four core burnout complaints. The psychometric qualities (validity and reliability) of the BAT are good, with a Cronbach’s alpha of 0.95 [66].

##### Work Engagement

Secondly, work engagement is measured through the Utrecht Work Engagement Scale (UWES-9) [67]. The UWES-9 includes 9 items measuring three scales, answered on a 7-point Likert scale (1 = never, 7 = daily): vigor (e.g., “At my work, I feel bursting with energy”), dedication (e.g., “I am enthusiastic about my job”), and absorption (e.g., “I am immersed in my work”). An average overall work engagement score is calculated based on the three scales, given the demonstrated acceptable one-dimensional construct [67]. The psychometric qualities of the UWES are good. The Cronbach’s alpha for the average overall work engagement score ranges from 0.85 to 0.94 across various studies [67].

##### Job Satisfaction

Finally, job satisfaction is measured by three items of the medical doctor version of the Leiden Quality of Work Questionnaire (LQWQ) [68]. The medical doctor version of LQWQ is comparable to the LQWQ for nurses (LQWQ-N) [69,70,71]. Responses are rated on a 4-point Likert scale, ranging from 1 (totally disagree) to 4 (totally agree). An example item is: “If I had to choose now, I would take this job again.” The LQWQ [68] and LQWQ-N [69,70] have been demonstrated to be reliable and valid.

#### 2.6.3. Secondary Outcome Measures

The secondary outcomes in this study are turnover intention, OSE, and perceived impact on health/well-being.

##### Turnover Intention

First, turnover intention is assessed through three items of the medical doctor version of the LQWQ [68]. An example item is: “I intend to search for a job outside this organization within the next 3 years.” Responses are answered on a 4-point Likert scale (1 = totally disagree, 4 = totally agree).

##### Occupational Self-Efficacy

Second, OSE is measured by the short version (6 items) of the Occupational Self-efficacy Scale (OSS-SF) [72]. An example item is: “Whatever comes my way in my job, I can usually handle it.” The items are answered on a 6-point Likert scale (1 = not at all true, 6 = completely true). The psychometric qualities of the OSS-SF have been demonstrated to be good (i.e., Cronbach’s alpha ranged between 0.85 and 0.90).

##### Perceived Impact on Health/Well-Being

Third, perceived impact on health/well-being is measured with a self-developed item: “Are the initiated changes affecting your health/well-being?” The response choices are “yes”, “no”, or “no changes have been initiated yet”. An additional item is presented when participants indicate that initiated changes affect their health/well-being: “To what extent is the impact of the initiated changes on your health/well-being positive or negative?”, rated on a 7-point Likert scale (1 = very negative, 7 = very positive). In contrast to the other outcomes, this secondary outcome is not measured at the pre-test, only at the intermediate evaluations and post-test.

#### 2.6.4. Mediators

Various mediators are included to evaluate whether the effect of the intervention on the primary and secondary outcomes is mediated by changes in PSC, job characteristics (job demands and job resources), and perceived impact on the work situation.

##### Psychosocial Safety Climate

First, PSC is measured by the demonstrated valid and reliable four-factor Psychosocial Safety Climate Survey (PSC-12), added with a fifth factor (group norms and behavior) and differentiation between management layers (top management and direct supervisor) [28,29,73,74]. This scale consists of five subscales, consisting of 15 items: top management (e.g., “Senior management considers employee psychological health to be as important as productivity”), direct supervisor (e.g., “My direct supervisor clearly considers the psychological health of employees to be of great importance”), group norms and behavior (e.g., “In our workplace, we remind each other of the rules and regulations regarding psychological stress”), communication (e.g., “My complaints, remarks and contributions to resolving psychological stress in the organization are listened to”), and participation and involvement (e.g., “In my organization, the prevention of psychological stress involves all levels of the organization”). The items are rated on a 5-point Likert scale, ranging from 1 (totally disagree) to 5 (totally agree). An average overall PSC score is calculated based on the five subscales.

##### Job Demands

Second, job demands (5 sub-scales) are assessed through the medical doctor version of the LQWQ [68], extended with items based on the Demand-Induced Strain Compensation Recovery Questionnaire (DISQ) [75], and the Questionnaire on the Experience and Evaluation of Work (QEEW) [76]. The following job demands are measured, rated on a 4-point Likert scale (1 = totally disagree, 4 = totally agree): time pressure (6 items), emotional workload (4 items), cognitive workload (5 items), physical workload (5 items), and social harassment (4 items).

##### Job Resources

Third, job resources (12 sub-scales) are also measured by the medical doctor version of the LQWQ [68] (rated on a 4-point Likert scale (1 = totally disagree, 4 = totally agree)), and a 6-item measure to assess team reflexivity (answered on a 5-point Likert scale (1 = totally disagree, 5 = totally agree)) [77]. The following job resources are measured: autonomy (4 items), recovery within worktime (4 items), social support from supervisor (4 items), social support from colleagues (4 items), work procedures (4 items), role clarity (4 items), opportunities for development (4 items), staffing levels (5 items), equipment and materials (3 items), internal communication (4 items), (financial) rewards (6 items), and team reflexivity (6 items).

##### Perceived Impact on the Work Situation

Fourth, perceived impact on the work situation is measured with a self-developed item: “Are the initiated changes affecting your work situation?”. The response choices are “yes”, “no”, or “no changes have been initiated yet”. An additional item is presented when participants indicate that initiated changes affect their work situation: “To what extent is the impact of the initiated changes on your work situation positive or negative?”, rated on a 7-point Likert scale (1 = very negative, 7 = very positive). In contrast to all mediators, this mediator is not measured at pre-test, only at the intermediate evaluations and post-test.

#### 2.6.5. Process Factors

Barriers and facilitators to the intervention’s effectiveness are measured by process factors at the intermediate evaluations and post-test. Items are based on the Intervention Process Measure (IPM) [48], and a process evaluation checklist [78]. The process factors that are measured are the degree of actual implementation of changes, information provision, medical doctors’ involvement, management support, and medical doctors’ mental models. All items are answered on a 7-point Likert scale, ranging from 1 (not at all) to 7 (to a very high degree). First, examples of changes or actions that have been initiated in the past intervention phase are provided. Next, the degree of implementation of changes is measured with one item: “Such changes have been initiated in our organization in the past months”. Information provision is measured with the item: “I am informed on the progress of such changes/actions’’. Medical doctors’ involvement is measured with three items (e.g., “I have the opportunity to give my views about the changes before they are implemented”). Management support is measured with three items (e.g., “The management of my organization supports the medical doctors in this change process”). Medical doctors’ mental models consist of appraisals of focus and approach of the intervention and positive expectations. Appraisals of focus and approach of the intervention are measured with two items: “How satisfied are you generally with the focus of the initiated changes?” and “How satisfied are you generally with the way the changes are being initiated?”. Lastly, positive expectations are measured with five items (e.g., “I expect that the changes will lead to a situation in which I can work in a healthier and safer manner”).

### 2.7. Statistical Analyses

#### 2.7.1. Sample Size

A simulation study was conducted to calculate power given our sample size. In this simulation study, a fixed effect of time (i.e., the intervention effect) and random intercepts with respect to group and participant were estimated. Furthermore, in half of the simulated datasets, the effect of time was constant across groups and participants, and in half of the datasets the strength of the effect of time varied between groups. The simulation assumed a total sample size of 514 medical doctors, distributed over 24 groups, measured at four measurement points. The results demonstrated there is about 80% power to detect effect sizes between 0.10 and 0.25, depending on attrition and heterogeneity between groups and participants. With a higher intraclass correlation coefficient (ICC), lower attrition, and lower effect heterogeneity, the study can detect smaller effects. An additional file demonstrates the simulation study in more detail (see Appendix A).

#### 2.7.2. Quantitative Evaluation

In this study, we use linear mixed-effect models (LMMs), to evaluate the effect of the intervention. LMM is suitable due to the three-level hierarchical structure: level 1 (time: measurement points), level 2 (participant: medical doctors), and level 3 (group: medical doctor groups). Moreover, LMM allows for variation in the number of observations per measurement point. This is important because the number of participants in each group may vary over time (T1-T2-T3-T4). We will only include medical doctors who have completed at least two measurement points to capture changes over time. First, to assess the effect of the intervention on the primary and secondary outcomes (Hypothesis 1) from pre-test (T1) to post-test (T4), and its intermediate effects (T1-T2-T3-T4), we will perform a linear mixed-effect model. We will include a fixed effect of time and random intercepts for the group and participants. We will also apply a random slope for the group to quantify the heterogeneity of the intervention unless this heterogeneity is not substantial enough to be estimated well. Pairwise post hoc testing using the Tukey method will detect where the effects occur between the measurement points. The intervention will be perceived as successful if at least one SE indicator (i.e., burnout complaints, work engagement, or job satisfaction) improves significantly. Second, to assess whether changes in PSC, job characteristics, and perceived impact on the work situation mediate the effect of the intervention on the primary and secondary outcomes (Hypothesis 2), we will perform a multilevel mediation analysis. Third, to assess whether process factors moderate the effectiveness of the intervention over time (Hypothesis 3), we will perform linear mixed-effect models per potential moderator (process factor) in combination with the mediators, primary and secondary outcomes. In this case, we will also include random intercepts for the group and participant, but now we will use fixed effects for the interaction between time and moderator. In the case of a significant result, we will use the Tukey method to indicate between which measurement points the moderation effect occurred.

## 3. Discussion

This study protocol presents the design of an organizational-level workplace intervention aimed at enhancing SE (lowering burn-out complaints, and enhancing work engagement and job satisfaction). In this study, we will implement a two-year-long organizational-level workplace intervention using the Participatory Action Research (PAR) approach among 24 groups of Dutch medical doctors. These groups are the key agents of change in the intervention. A two-year duration, including intermediate evaluations, is chosen, because system changes at work take time to develop and implement, which enables achievement of positive changes in PSC or job characteristics, and impacts SE [79]. Given that the evaluation of organizational-level workplace interventions is challenging [46,53,56], both the effect and process will be evaluated. More specifically, we will examine whether changes in PSC, job characteristics (job demands and job resources), and/or perceived impact on the work situation, can explain changes in SE. Further, several relevant process factors (i.e., degree of implementation, information provision, medical doctors’ involvement, management support, and medical doctors’ mental models) that are important to the success of such organizational-level workplace interventions will be investigated. Thus, the proposed study provides insight into if, why, for whom, and under which circumstances an organizational-level workplace intervention using the PAR approach works, which is essential to strengthen and refine theories and maximize the effectiveness of future interventions [46,53,56].

### 3.1. Limitations

Despite the above-mentioned strengths, this study also has some limitations. The first limitation is the absence of a control group, while randomized control trials (RCTs) are seen as the golden standard for evaluating intervention effectiveness. However, research also indicates that RCT is not always feasible or suitable for complex organizational interventions [53]. This study therefore examines the overall effect of the intervention, how the intervention creates changes, and the conditions under which the intervention is the most effective [53]. The second limitation, related to the first limitation, is that it can be difficult to evaluate the effectiveness of the intervention given challenges that can emerge during the implementation of the intervention in a real-world setting, which may impact intervention effectiveness [45,46,47,48,53,56]. This point also stresses the importance of the process evaluation to understand how contextual factors facilitate or hinder intervention effectiveness. The last limitation is that the groups subscribed themselves to the intervention and were self-motivated to improve their work situation, resulting in selection bias. Hence, the included groups may experience worse PSC, job characteristics, and/or SE. To the extent feasible, we will compare our baseline data with data available in the literature on medical doctors (i.e., the prevalence of burnout complaints or work engagement) to determine the potential influence of selection bias. Furthermore, some groups subscribed themselves to the intervention when the COVID-19 pandemic played a major role in healthcare. Given the nature of organizational-level workplace interventions, it was not possible to control such significant global events, which may have influenced the course of the intervention and may impact the results of the intervention. We will interpret the findings taking this context into account, and will exploratorily examine whether the effect of the intervention on the SE indicators differs between medical doctor groups that started earlier during the COVID-19 pandemic and those that started after its impact subsided.

### 3.2. Implications for Theory and Practice

This study is theoretically relevant as it empirically tests the JD-R model extended with PSC. As such, it will give insight into whether this model can explain medical doctors’ SE. Furthermore, this study enriches our understanding of relevant boundary conditions for achieving positive changes in the work environment through organizational interventions.

This study is also relevant for practice, given the increasing concerns related to SE among medical doctors. By providing a deeper understanding of the role of PSC and job characteristics in organizational-level workplace interventions, this study can provide organizations with information on how to structurally address SE in medical doctors. Furthermore, if the intervention proves to be effective, this will indicate that organizational-level workplace interventions with a PAR approach are a feasible method for addressing SE in medical doctors. Lastly, the evaluation of the intervention process will provide insight into which factors are important for intervention success. This knowledge can be used in the design and implementation of future organizational-level workplace interventions to optimize their success, and ultimately contribute to improved SE for medical doctors and employees in general.

## 4. Conclusions

The implementation and evaluation of organizational-level workplace interventions using a Participatory Action Research approach is limited among medical doctors. This study’s results will provide insights into the effectiveness of the intervention in improving sustainable employability (SE), examine whether this improvement is mediated by changes in psychosocial safety climate and/or job characteristics, and identify which facilitators or barriers in the implementation process contribute to its outcomes. The knowledge gained can inform the design and implementation of future organizational-level workplace interventions aimed at enhancing SE.

## Figures and Tables

**Figure 1 ijerph-21-01561-f001:**
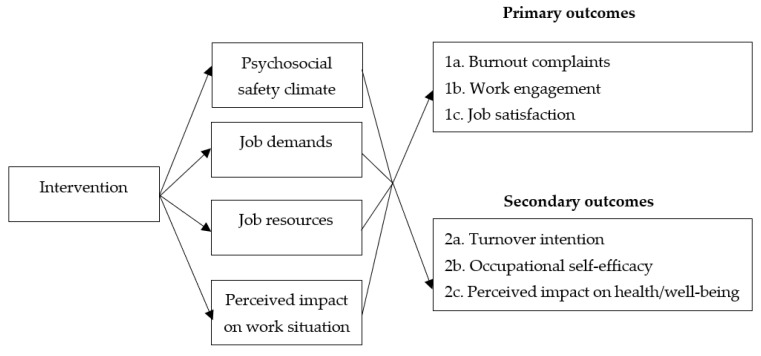
Conceptual model of study.

**Figure 2 ijerph-21-01561-f002:**
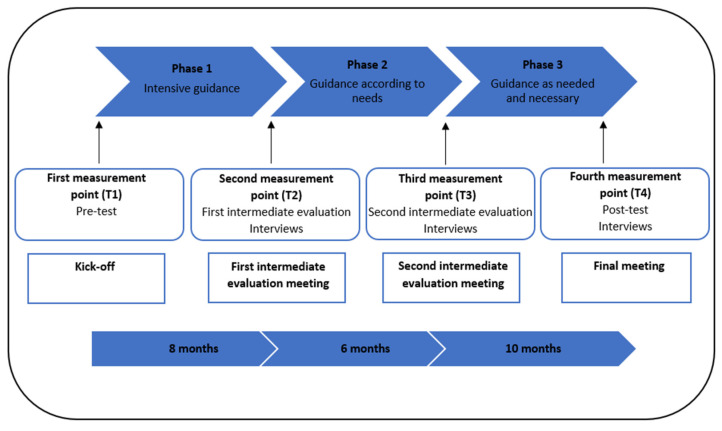
Study design protocol.

**Table 1 ijerph-21-01561-t001:** Overview of the primary and secondary outcomes, mediators, and process factors per measurement point.

	Pre-Test	First Intermediate Evaluation	Second Intermediate Evaluation	Post-Test
**Primary outcomes**				
Sustainable employability				
Burnout complaints	X	X	X	X
Work engagement	X	X	X	X
Job satisfaction	X	X	X	X
**Secondary outcomes**				
Turnover intention	X	X	X	X
Occupational self-efficacy	X	X	X	X
Perceived impact on health/well-being		X	X	X
**Mediators**				
Psychosocial safety climate	X	X	X	X
Job resources				
Autonomy	X	X	X	X
Within-worktime recovery	X	X	X	X
Social support supervisor	X	X	X	X
Social support colleagues	X	X	X	X
Work procedures	X	X	X	X
Role clarity	X	X	X	X
Development opportunities	X	X	X	X
Staffing levels	X	X	X	X
Equipment and materials	X	X	X	X
Internal communication	X	X	X	X
(Financial) reward	X	X	X	X
Team reflexivity	X	X	X	X
Job demands				
Time pressure	X	X	X	X
Emotional workload	X	X	X	X
Cognitive workload	X	X	X	X
Physical workload	X	X	X	X
Social harassment	X	X	X	X
Perceived impact on work situation		X	X	X
**Process factors**				
Degree of implementation		X	X	X
Information provision	X	X	X
Medical doctors’ involvement	X	X	X
Management support	X	X	X
Medical doctors’ mental models			
Appraisal intervention focus and approach	X	X	X
Positive expectations	X	X

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
