# Peer review of "An Organizational-Level Workplace Intervention to Improve Medical Doctors’ Sustainable Employability: Study Protocol for a Participatory Action Research Study"

_ijerph, 2024, doi:10.3390/ijerph21121561_

Round 1
Reviewer 1 Report
Comments and Suggestions for Authors
Paper studies Compromised Sustainable Employability (SE) of medical doctors on a large sample. The empircial study posits significant research questions and is able to answer them.
The introduction sets the background well, but the hypotheses section lacks support, so some of the parts of intro can be used to support hypotheses, and also an underlying theory of JD r was used , to base the model. But COR theory can also be applied. The study is also linked to studies such as: Enhancing Job Performance: The Critical Roles of Well-Being, Satisfaction, and Trust in Supervisor.
The discussion can be more detailed, and structured more traditionally without strenghts section.
Author Response
Dear reviewer 1,
Your suggestions were helpful, and we would like to express our gratitude for your careful reading and ideas to improve the manuscript.
Please see the attachment for a point-by point response to your comments.

Reviewer 2 Report
Comments and Suggestions for Authors
1.The article presents a well-structured and ambitious research protocol that addresses a significant and relevant issue: the sustainable employability of medical doctors.
2. The introduction lacks a clear statement of the research objective. It is only mentioned on page 4: "Therefore, to fill this research gap, the present study aims to test a two-year-long organizational-level workplace intervention using a PAR approach to enhance SE via changes in PSC and job characteristics in groups of medical doctors (see Figure 1)." Additionally, the introduction implies that research results will be presented, which is not the case. It should be clarified that the article serves as a preparation for the publication of findings from research that has already taken place.
3. The research project is broad in scope, and the scale and duration of the study are indeed impressive. However, presenting such a large-scale study with seven hypotheses may seem overly ambitious for a single scientific article. It might be worth considering reducing the number of hypotheses or limiting the variables. Alternatively, a thorough revision of the article could help simplify and enhance the clarity of the research description. This would ensure the study is presented in a more concise and focused manner.
4. The research tools, methodologies, and proposed methods of statistical analysis in the indicated article are appropriately selected. However, it may still be beneficial to review the article for potential simplifications or improvements in the clarity of the presentation, as previously mentioned.
5. The research results are not clearly explained. The article serves as a research protocol without presenting actual findings, and only simulated observations are provided in the appendix. It would be beneficial to clearly state at the beginning of the article that it focuses on the preparation for the research. In the results section, the expected outcomes could be further elaborated. Additionally, earlier sections could include a more extensive review of other authors' findings to provide greater context and support for the research approach.
6. It would be beneficial to expand the literature review to include works published after 2020. This would ensure that the article reflects the most current research and developments in the field.
7. In the article, it would be worthwhile to address the impact of the COVID-19 pandemic, particularly among healthcare professionals. The pandemic significantly affected various aspects of both professional and personal life, which may be crucial in the context of the study conducted between 2020 and 2022—during the pandemic period. Understanding this context will provide a better illustration of the research findings and their implications.
Author Response
Dear reviewer 2,
Your suggestions were helpful, and we would like to express our gratitude for your careful reading and ideas to improve the manuscript.
Please see the attachment for a point-by point response to your comments.

Reviewer 3 Report
Comments and Suggestions for Authors
I believe that the topic addressed by the authors may be of interest depending on the results obtained. The introduction is well elaborated and correctly defines the variables dealt with in the study and their justification. The objectives and hypotheses are also correct. The proposed methodology seems adequate as long as it specifies more clearly the specific actions in the intervention by the agents involved.
The main problem I find is that there are no results because the study is not carried out; what the authors present is a project. I do not know to what extent the journal accepts the publication of research-intervention projects or protocols. I consider that if they are going to obtain the final results by December 2024, they should wait to publish the research with the results.
Since there are no results, there is no discussion or conclusions based on them.
For all the above reasons, I think you should consider whether a research project or protocol can be published.
Author Response
Dear reviewer 3,
Your suggestions were helpful, and we would like to express our gratitude for your careful reading and ideas to improve the manuscript.
Please see the attachment for a point-by point response to your comments.

Round 2
Reviewer 3 Report
Comments and Suggestions for Authors
I believe that with the changes made by the authors and the clarification that the publisher accepts protocols, they are sufficient for this article to be published.